# Keep the Cost Down: A Review on Methods to Optimize LLM's KV Cache Consumption

**Shi Luohe & Zhang Hongyi**
National Engineering Research Center
for Multimedia Software,
School of Computer Science,
Wuhan University,
Wuhan, 430072, P. R. China
{shiluohe, harryzhang}@whu.edu.cn

**Yao Yao**
Department of Computer Science
and Engineering,
Shanghai Jiao Tong University
yaoyao27@sjtu.edu.cn

**Li Zuchao**[*]
National Engineering Research Center
for Multimedia Software,
School of Computer Science,
Wuhan University,
Wuhan, 430072, P. R. China
zcli-charlie@whu.edu.cn

**Zhao Hai**
Department of Computer Science
and Engineering,
Shanghai Jiao Tong University
zhaohai@cs.sjtu.edu.cn

## Abstract

Large Language Models (LLMs), epitomized by ChatGPT's release in late 2022, have revolutionized various industries with their advanced language comprehension. However, their efficiency is challenged by the Transformer architecture's struggle with handling long texts. KV Cache has emerged as a pivotal solution to this issue, converting the time complexity of token generation from quadratic to linear, albeit with increased GPU memory overhead proportional to conversation length. With the development of the LLM community and academia, various KV Cache compression methods have been proposed. In this review, we dissect the various properties of KV Cache and elaborate on various methods currently used to optimize the KV Cache space usage of LLMs. These methods span the pre-training phase, deployment phase, and inference phase, and we summarize the commonalities and differences among these methods. Additionally, we list some metrics for evaluating the long-text capabilities of large language models, from both efficiency and capability perspectives. Our review thus sheds light on the evolving landscape of LLM optimization, offering insights into future advancements in this dynamic field. Links to the papers mentioned in this review can be found in our Github Repo https://github.com/zcli-charlie/Awesome-KVCache.

## 1 Introduction

Since the release of ChatGPT, Large Language Models (LLMs) are gradually having a profound impact on people's lives and are standing out in various fields (Wu et al., 2023; Roumeliotis & Tselikas, 2023b;a). These LLMs face a computational challenge: their Decoder-Only Transformer architecture has a quadratic time complexity when processing text sequences. During inference, the auto-regressive decoding mechanism amplifies this issue, as it repeats the process for each token generated. KV Cache, by storing the keys and values tensor in attention module generated by past tokens, can reduce the time complexity required to generate each token to linear, greatly improving inference efficiency. However, the use of KV Cache is not all advantageous. KV Cache will increase linearly with the length of

---

[*]Corresponding author

the sequence, and the memory required will become larger and larger, especially for giant models like GPT-3 (Floridi & Chiriatti, 2020). Moreover, each individual dialogue needs to save its own KV Cache, and different dialogues can hardly reuse them, which will become a bottleneck in the generation speed on modern inference hardware like GPU as they usually suffers from the low memory bandwidth comparing to their computing speed (Yu et al., 2022).

Recent months have seen emerging work on optimizing KV Cache, making it a critical focus for enhancing LLMs' performance with longer contexts. This review presents various methods of KV Cache optimization, clarifying their interrelationships and comparing their core ideas. We discuss extensive methods to reduce memory space: before inference, the model itself can be compressed, or the architecture can be changed, completely abandoning attention with quadratic complexity; during inference, from the model input prompt level, to the embedding level, and then to the KV Cache level, compression can be performed. This paper primarily focuses on editing, modifying, and optimizing KV-Cache itself, with other methods briefly mentioned in Appendix C. These optimizations are considered the safest, most effective, and compatible approach known to date. A general preview can be found in Figure 1.

This review unfolds in chronological order of the LLMs, from the training phase, to the deployment phase, and finally to the post-training phase. In the **training stage**, we will introduce the KV Cache compression methods that can be used during model pre-training. These methods are usually the most effective, but they are not suitable for modifying existing models or scenarios with low computational budget. In the **deployment stage**, we will introduce the use of different frameworks to optimize the use of KV Cache. The methods in this section will not make a large number of modifications to KV Cache itself, but can significantly optimize its efficiency in the same environment. Finally, in the **post-training stage**, we will introduce a large number of on-time optimization methods for KV Cache, mainly including three methods: Eviction, Merging, and Quantization.

Additionally, we introduce metrics to assess LLMs' performance on long texts, which are vital for KV Cache optimization. These metrics are divided into efficiency and performance aspects. Efficiency measures include model generation speed and space occupancy improvement, while performance metrics evaluate the impact on model capabilities, ensuring any capability loss remains acceptable.

In conclusion, this review chronologically introduces KV Cache optimization methods in LLMs, aiming to enhance model inference efficiency and context length. This review aims to summarize and analyze various LLM optimization methods from the perspective of KV-Cache, enabling the readers to better understand KV Cache optimization and build more effective, efficient, and sustainable LLMs.

## 2 Preliminary and Notations

Before introducing the KV Cache compression method, we must rigorously define frequently occurring symbols to avoid ambiguity in the narrative. The core function of LLM is to transform a natural number sequence input into equal amount of multiple probability distributions $\mathcal{LLM}(X) = \mathcal{LLM}(x_1 x_2 \ldots x_n) = p_1 p_2 \ldots p_n$. In this paper, we denote $X$ as the input sequence of integers, named tokens, with the length $n$, composed by $x_i$ as the $i$-th token. Then LLM translates them into $p_i$, which is the probability of the next token corresponding to $x_i$. In a LLM, the entire set of possible values for $x_i$ constitutes the vocabulary, $\mathcal{V}$. The size of the vocabulary is denoted as $V = |\mathcal{V}|$. Naturally, we have $X \in \mathcal{V}^n$, $x_i \in \mathcal{V} = \{0, 1, \ldots, V - 1\}$, and $p_i \in \mathbb{R}^V$. The input $X$ is translated from the natural language input PROMPT in tokenization, and usually $p_n$ is used to predict the next token with various sampling algorithms when auto-regression decoding.

Within the LLM, $L$ Transformer Decoder blocks are sandwiched between the Embedding layer and a linear layer (along with softmax). The Embedding layer translates every token $x_i$ into a $d$-dimensional vector $h_i^{(0)}$, while the linear layer and softmax transform every

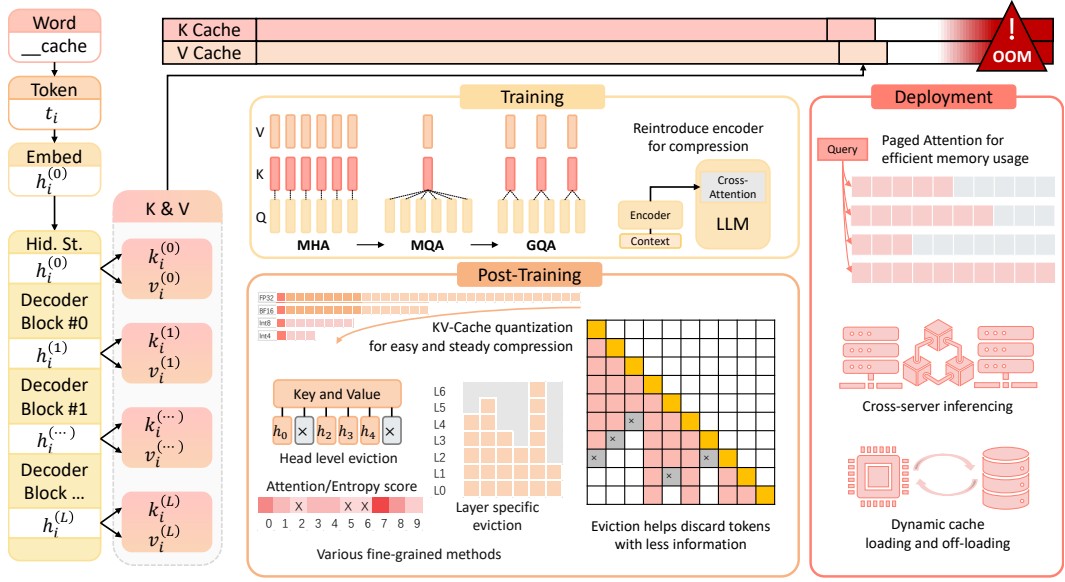

Figure 1: An overview of the main structure of KV Cache compression methods

$d$-dimensional vector $h_i^{(L)}$ into a probability distribution $p_i$. We denote $L$ as the layer count, and $h_i^{(l)}$ as the hidden state of $i$-th token, after $l$-th layer (here we treat the embedding layer as the 0-th layer). At last, $H^{(l)}$ represents the matrix formed by concatenating $n$ $h_i^{(l)}$ vectors.

Each Transformer Decoder Block consists of two parts: Self-Attention and Feed-Forward Network (FFN). Each part has its own residual connection and Normalization (Norm) operation. In this paper, we only focus on the Self-Attention part. In the Decoder Block, each $h_i^{(l)}$ is mapped to three new vectors $q_i^{(l)}$, $k_i^{(l)}$, and $v_i^{(l)}$ using three trainable matrices $W_q^{(l)}$, $W_k^{(l)}$, paired with position embedding matrix $R_i$, and $W_v^{(l)}$. In other words, $H^{(l)}$ is mapped to $Q^{(l)}$, $K^{(l)}$, and $V^{(l)}$. Another matrix $W_o^{(l)}$ is then used to map the output of the Self-Attention layer to the input $h'^{(l)}_i$ for a single token, and $H'^{(l)}$ for the entire sequence. The process is given by Formula 1.

$$H'^{(l)} = \text{softmax}\left(\text{mask}\left(\frac{Q^{(l)} \cdot K^{(l)\mathrm{T}}}{\sqrt{d}}\right)\right) \cdot V^{(l)} \cdot W_o^{(l)} \tag{1}$$

Modern LLMs utilize multi-head attention (MHA). The idea is to split $q$, $k$, and $v$ into $n_h$ smaller blocks, named heads, each with $d_h = d/n_h$ dimensions. For the $j$-th head of $l$-th layer, we denote $\{Q, K, V, W_q, W_k, W_v, q_i, k_i, v_i\}^{(l,j)}$ for the corresponding partition of these matrices or vectors. In MHA, Causal Mask $\text{mask}(\cdot)$ was used to prevent earlier tokens attend on later ones, by adding $-\inf$ to the upper right triangle of the pre-softmax attention matrix. This crucial property ensures that the $K$ and $V$ computed by preceding tokens will not be affected by subsequent tokens. Therefore, for each token newly added during auto-regression decoding, we only need to update previous $K$ and $V$, as $\{K^{(l)}, k_n^{(l)}\} \longrightarrow K^{(l)}$ and $\{V^{(l)}, v_n^{(l)}\} \longrightarrow V^{(l)}$, and perform Formula 2. The $K$ and $V$ which we retained are called **KV Cache**.

$$h'^{(l)}_n = \text{Concatenate}_{j=1}^{n_h}\left(\text{softmax}\left(\text{mask}\left(\frac{q_n^{(l,j)} \cdot K^{(l,j)\mathrm{T}}}{\sqrt{d_h}}\right)\right) \cdot V^{(l,j)}\right) \cdot W_o^{(l)} \tag{2}$$

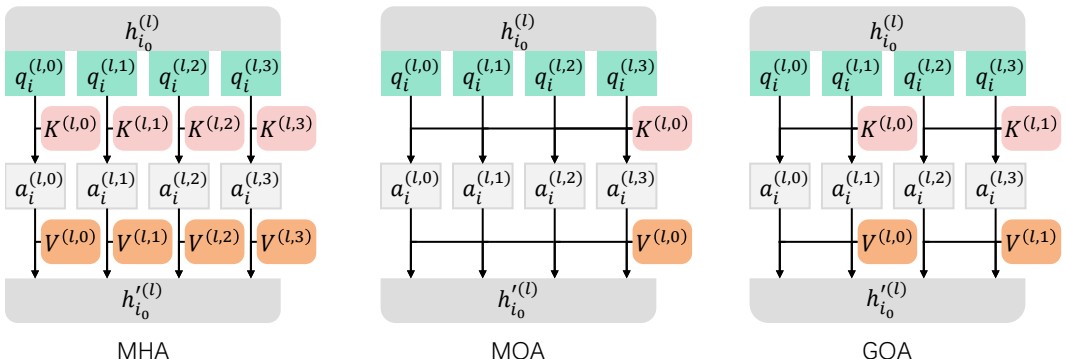

Figure 2: The comparasion between MHA, MQA and GQA. To be note that the final linear layer $W_o^{(l)}$ is not depicted here.

## 3 Training Stage Optimization

For LLMs that adopt the traditional Decoder-Only Transformer architecture, the most effective KV Cache compression method emerges during the pre-training phase. This is because, in this phase, the model possesses the greatest plasticity. The primary adjustment in this phase is to the model architecture, which, while still retaining the excellent properties of Attention, reduces the size of the generated Keys and Values vectors to a quarter or even less.

Shazeer (2019) proposed Multi-Query Attention (**MQA**) based on Multi-Head Attention (**MHA**). Shazeer claimed that even if we retain only one head for keys and values, we can still achieve satisfactory model performance. In this case, different query heads calculate attention scores with the same key head. Although there is only one head left for value, it will receive $n_h$ different combination weights, resulting in $n_h$ combinations. Doing so can instantly optimize the KV Cache space usage to $1/n_h$ of the original.

Reducing the number of keys and values heads from $n_h$ to 1 is undoubtedly an aggressive strategy. Hence, Ainslie et al. (2023) proposed Grouped Query Attention (**GQA**), a method that can better balance speed and performance. In GQA, query heads are divided into $n_g$ groups. Each group shares one key head, serving as the $n_h/n_g$ different combination weights for the corresponding value head. Ultimately, the KV Cache we need to store will be reduced to $n_g/n_h$. Compared to MHA and MQA, GQA introduces an adjustable parameter $n_g$. When $n_g = 1$, we get MQA; when $n_g = n_h$, we get MHA. When $n_g$ is between 1 and $n_h$, efficiency and performance achieve a more granular balance. Moreover, GQA can save a substantial number of parameters within the Attention module, with the ratio of savings $\eta = 0.5 + 0.5 \cdot n_g/n_h$. A comparison of MHA, MQA and GQA can be found at Figure 2. The current usage of GQA and MQA in open-source models can be found in Appendix A.

If GQA reuses the KV Cache within layers, then the method proposed by Sun et al. (2024); Brandon et al. (2024); Goldstein et al. (2024) involves inter-layer KV Cache reuse. By sharing KV headers across layers, we can reduce the KV Cache size while maintaining model capacity. Specifically, the Brandon et al. (2024) method directly reuses these contents across layers, as **CLA**, whereas **YOCO** and **GoldFinch** in Sun et al. (2024); Goldstein et al. (2024) introduce more significant changes: they generate the all layers' KV Cache using linear models, like RetNet (Sun et al., 2023a) and RWKV (Peng et al., 2023a). However, it's essential to note that these methods do not optimize the memory bandwidth bottleneck of the KV Cache. Cross-layer reuse results in scattered access to the same KV Cache, and only meticulously crafted CUDA kernels can effectively reduce data exchanges between different cache levels, if such reduction is possible at all.

We can consider reuse as a form of low-rank compression, and complete low-rank compression should have even greater potential. The method proposed by DeepSeek-AI et al. (2024), called Multi-Head Latent Attention (**MLA**), exemplifies this. MLA achieves KV

Cache decompression by utilizing an additional dimension-expanding matrix, rather than merely reusing the existing content. However, it's essential to note that this method still does not address the memory bandwidth bottleneck, as the cache still needs to be expanded into large KV vectors.

Yen et al. (2024) proposed a new idea for the LLM architecture. **CEPE**, a framework that combines the pre-trained LLM with an Encoder module that serves as context compressor. In RAG scenario, the ultra-long context will greatly increase the need to store KV Cache. CEPE's additional Encoder compresses these reference texts, and then inputs them into the Decoder through cross-attention. The compressed text has a shorter sequence length than before, so the KV Cache is saved. This method can use the pre-trained LLM, but the added decoder and cross-attention layer still require a lot of extra training.

# 4 Deploy-Stage Optimization

From the perspective of inference systems, KV Cache presents significant challenges in two key areas. First, during the continuous $\{K, k_n\} \longrightarrow K$ and $\{V, v_n\} \longrightarrow V$ operations, memory is repeatedly allocated and released, resulting in substantial fragmentation. Second, the inability to batch-process KV Cache exacerbates severe memory bandwidth bottlenecks, leading to computational inefficiency. Addressing these two issues has become a central concern for inference systems.

Kwon et al. (2023) introduced the **Paged Attention** mechanism and the **vLLM** framework. Paged Attention draws on the page memory mechanism widely used in CPU memory, and uses an additional mapping table to map the KV Cache that used to be stored continuously to discontinuous GPU memory. During inference, in order to generate the next token, it is necessary to call KV Cache for calculation, and this process can also be efficiently completed through a set of custom CUDA kernels. The use of the Paged Attention mechanism results in almost no unused memory fragments and efficient inference.

Lin et al. (2024) further developed the idea of Paged Attention. Through the **DistAttention** it proposed and the **DistKV-LLM** built on it, KV Cache was able to achieve distributed deployment on multiple servers. This significantly improved the efficiency of providing LLM services using large-scale cloud servers.

Ye et al. (2024) aims to reuse KV Cache between different dialogues, achieving acceleration of the pre-fill stage and optimization of GPU memory occupancy. By establishing a dictionary tree for all historical dialogues, and finding the longest common prefix in the dictionary tree when a new dialogue request is received and reusing the corresponding KV Cache for this part, the **ChunkAttention** made the model avoid repeated calculation of some tokens in the pre-fill stage, speeding up the response speed of the deployment system. Moreover, Gao et al. (2024) extend this idea by introducing a hierarchical system that utilizes multiple storage devices.

Jin et al. (2024) and Lee et al. (2024) offload KV Cache to CPU with speculation. The core idea is to offload the primary portion of the KV Cache to the CPU, retaining only critical components. During inference, by speculatively reloading a small amount of KV Cache onto the GPU using the limited preserved information, we can save memory space on high-speed computational devices without compromising performance, all while avoiding extensive data exchanges.

He & Zhai (2024) introduces a novel approach: by collaboratively utilizing the CPU for attention computations, we can enhance GPU utilization efficiency, improve overall effective computational power, and accelerate inference speed.

Additionally, Qin et al. (2024) provides a comprehensive technical report from an enterprise perspective, optimizing the entire inference system.

# 5 Post-Training Optimizations

## 5.1 Eviction and Merging

**Eviction** methods are about the policies to discard unnecessary tokens. Two lines of approaches exist: static policies, which are designed before inference and remains consistent across every inference request, and dynamic policies, which utilize the information generated while inference to identify important tokens.

**Static policies** A straightforward approach for maintaining a fixed-size KV Cache is to retain recent tokens, known as slide window attention (Beltagy et al., 2020). However, the model will collapse when sequence length exceeds cache size (Xiao et al., 2023). Recently, Xiao et al. (2023) and Han et al. (2024) proposed that initial tokens almost consistently receive high attention weights across layers and heads. As a result, keeping KV Cache of both initial and recent tokens can maintain model performance for long contexts.

**Dynamic policies** Policies based on attention weights have been widely explored by researchers, as they believe that attention weights can provide insights into the importance of individual tokens. This can be leveraged in the design of dynamic strategies for eviction. Since it is impossible to predict which token will be necessary for future generation, the question arises whether it is possible to estimate the importance of a token based on history information.

Liu et al. (2023) empirically finds existence of **Repetitive Attention Pattern**. This pattern suggests that for two different tokens, there are similarities in what they are attending to and ignoring. So a natural hypothesis is that only the tokens which are important in previous steps will be significant in the future.

This provides us with the possibilities to estimate future significance of a token based on history information of attention weights. **Token Omission Via Attention (TOVA)** (Oren et al., 2024) proposes a simple greedy method to discard useless tokens. While keeping a fixed-length of KV-Cahce, **TOVA** evicts at each decoding step the tokens with minimal attention weights layer-wise.

**H2 Eviction Algorithm** (Zhang et al., 2023) uses accumulative normalized attention scores to decide which token to stay and at the same time keeps the recent tokens since they may show strong correlations with current tokens.

**PyramidInfer**(Yang et al., 2024a) uses a layer-wise approach with recent tokens occupying more weights. Apart from this, Yang et al. (2024a) believes that deeper layer contains more context redundancy, so it defines a decay ratio to shorten the length of KV Cache in deeper layers and thus forming a "pyramid".

However, while discarding unnecessary tokens seems to have minor influence on the original inference process, Adnan et al. (2024) observes that when evicting more tokens during the inference process, the distribution of normalized attention scores becomes uneven among the remaining ones. Therefore, **Keyformer**(Adnan et al., 2024) introduces additional methods to smooth the uneven distribution and approximate the original one when using full KV Cache.

**FastGen** (Ge et al., 2023) uses a hybrid strategy to optimize token selection for discarding. In prompt encoding phase, it selects the best policy for each head in KV Cache, and then uses these policies to decide which token to discard in decoding process. The policies include: a) keeping special tokens, b) keeping punctuation tokens, c) keeping recent tokens and d) keeping tokens with attention-weight-based policies.

**SparQ Attention** (Ribar et al., 2023) tries a different approach. Instead of reducing memory capacity, it aims at reducing the amount of data transferred. It first uses the norm of $q_i^{(l)}$ to decide which $k_i^{(l)}$ needs to be fetched, and approximate the attention scores based on this subset of queries and keys. Then it selects the top-$K$ attention weights and fetch their corresponding Key-Value pair to calculate the output. To compensate the values that are

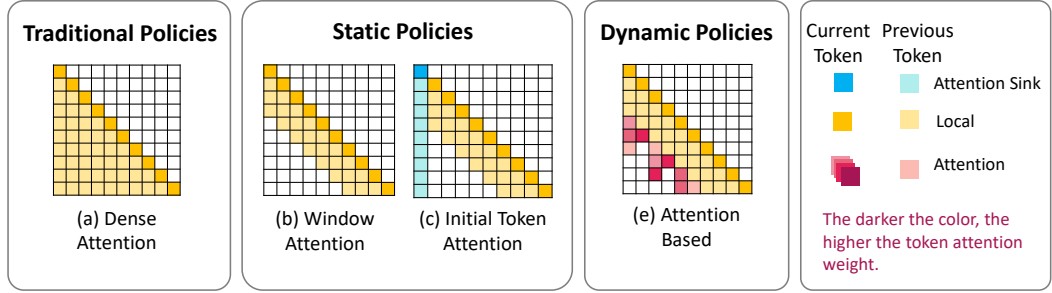

Figure 3: The comparison between different KV Cache policies

considered insignificant, it keeps a running mean of value vectors and interpolates between this running mean and the output of attention so as to approximate the original distribution.

Methods based on attention score also includes Corallo & Papotti (2024); rae Jo & Shin (2024); Xu et al. (2024); Fu et al. (2024).

While attention scores provide ample information, they are incompatible with certain optimization methods, such as Flash Attention (Shah et al., 2024). Therefore, investigating how to evict tokens without relying on attention scores is a novel research topic. Several approaches address this challenge. Tang et al. (2024) leverages the prior-known properties of attention heads to determine whether to evict. Yao et al. (2024) uses token entropy to decide whether to remove tokens. Devoto et al. (2024) relies on the $L_2$ norm of keys to determine token eviction. All these methods can be combined with Flash Attention. Notably, Devoto et al. (2024)'s approach demonstrates excellent granularity and controllability.

**Merging**, as a relatively recent method, represents a natural extension of Eviction, transitioning from hardmax to softmax. The current key challenges in this area include determining the merging region and selecting an appropriate scheme for combining keys and values.

Nawrot et al. (2024) proposed Dynamic Memory Compression (**DMC**), which use a single, after-trained dimension in key to determine whether to merge or not, and use attention score as the reference of merging. Wang et al. (2024) proposed the **KVMerger**, to merge KV by Gaussian weights and attention score.

Yu et al. (2024) proposed a novel approach to merge KV Cache of each token across different attention head, with a small amount of additional training, eventually turning an MHA model into a GQA model. Also requires additional training, Pang et al. (2024) proposed Anchor-LLM, which utilizes an innovative anchor-based self-attention, to help model learn how to merge KV Cache into a few token when training, achieving an outstanding performance.

## 5.2 Quantization

Another commonly utilized method for compressing Key-Value (KV) cache is through quantization. This approach effectively compresses data by mapping tensor values, originally in full precision, to discrete levels and storing them at a reduced precision. There are two primary categories of KV Cache quantization: Full quantization and KV Cache-only quantization. Full quantization involves compressing both the model weights and the KV Cache, thereby reducing the memory footprint of the entire model. On the other hand, KV Cache-only quantization specifically targets the KV Cache activations, selectively compressing them to conserve memory while leaving the model weights in their original state. This tailored approach can offer a balance between model efficiency and performance.

**KV Cache only quantization** Hooper et al. (2024) further identified that Key matrices in language models often exhibit distinct outlier channels characterized by larger average magnitudes compared to others. To address this, They proposed KVQuant, a method

to quantize Key and Value activations using distinct strategies. KVQuant's key features include: (1) Per-Channel and Per-Token Quantization, employing channel-based quantization for Keys and token-based for Values, effectively managing outlier distributions and mitigating distortion from Rotary Positional Embeddings (RoPE). (2) Sensitivity-Weighted Non-Uniform Datatypes, which enable more accurate activation distribution representation within layers. (3) Isolation of Outliers, minimizing skewness in quantization ranges to enhance quantization precision. (4) Normalization of Quantization Centroids, aligning post-quantization distribution mean and standard deviation with pre-quantization values, beneficial in ultra low-bit quantization like 2-bit scenarios.

| Model | Method | GSM8k | MMLU | BBH | HellaSwag | ARC-Challenge | WinoGrande | WikiText2 | PTB | C4 |
|-------|--------|-------|------|-----|-----------|---------------|------------|-----------|-----|-----|
| | | | | | ACC | | | | ppl | |
| LLaMA2-7B | FP16 baseline | 16.30 | 33.58 | 44.80 | 56.67 | 39.84 | 67.24 | 5.47 | 37.91 | 7.26 |
| | GEAR | 15.70 | 33.01 | 44.45 | - | - | - | - | - | - |
| | WKVQuant | - | - | - | 56.14 | 40.78 | 67.48 | 5.64 | 38.85 | 7.49 |
| | QAQ | - | - | - | 76.30 | - | - | - | - | - |
| LLaMA2-13B | FP16 Baseline | 30.34 | 40.79 | - | 59.69 | 45.56 | 69.69 | 4.88 | 50.93 | 6.72 |
| | GEAR | 27.97 | 37.38 | - | - | - | - | - | - | - |
| | WKVQuant | - | - | - | 58.98 | 43.94 | 68.75 | 5.00 | 52.36 | 6.89 |
| | QAQ | - | - | - | 76.60 | - | - | - | - | - |

Table 1: Comparison of different quantization methods. Results are reported by Hooper et al. (2024), Dong et al. (2024b) and Dong et al. (2024b)

Dong et al. (2024a) introduces LESS (Low-rank Embedding Sidekick with Sparse policy), which integrates a constant-sized cache with eviction-based cache methods (Section 5). Inspired by recurrent networks, LESS utilizes a constant-sized low-rank cache by replacing the softmax with a separable similarity metric calculated by learnable row-wise functions. By doing so, LESS obtains the ability to accumulate history token information before they are discarded from KV Cache, allowing for continued access to previous information.

Yang et al. (2024b) proposd Mixed-precision KV Cache (MiKV), a reliable cache compression method that retains evicted KV pairs in reduced precision to preserve information and maintains important KV pairs in higher precision to ensure generation quality. MiKV employed Dynamic Outlier Awareness to dynamically balance the outliers manifested in the query and keys to reduce quantization error. Dynamic Outlier Awareness multiplies and divides a channel balancer to the keys and queries to mitigate the impact of outliers.

Dong et al. (2024b) proposed the Quality Adaptive Quantization (QAQ). QAQ utilized separate quantization strategies for key caches and value caches, ensuring that the key cache, which is more sensitive, is quantized in a way that less affects model performance. QAQ also uses an attention window to predict future attention scores based on the history of attention values. This ensures that the quantization does not overly compress tokens that may become more important in subsequent generation steps.

Kang et al. (2024) proposed GEAR (Generative Inference with Approximation Error Reduction) which integrates three techniques: (1) Uniform quantization for the majority of entries. (2) Low-rank matrix approximation for quantization residuals. (3) Sparse matrix to handle errors from outlier entries.

**Full quantization** Sheng et al. (2023) proposed FlexGen which compresses both model weights and attention cache to 4 bits without significant accuracy loss.

Yue et al. (2024) took FlexGen a step further and proposed WKVQuant. For weight quantization, WKVQuant ultize the OmniQuant (Shao et al., 2023). For KV Cache quantization, WKVQuant employs following three strategies: Past Only Quantization, Two-dimensional Quantization and Cross-block Reconstruction Regularization. Different from KVQuant (Hooper et al., 2024) which focus on calibrating for per-channel(token) thresholds to ignore outlier, WKVQuant focuses on aligning and smoothing each channel and token.

## 6 Evaluation

In this section, we first introduce the datasets commonly used in KV Cache optimization, followed by a presentation of frequently used evaluation metrics.

### 6.1 Datasets

**Long context benches**   Currently, there are several benchmarks that rely on extracting information from extremely long texts to answer questions. These benchmarks typically require models to read a substantial reference document, often containing 4,000 to 20,000 tokens, and then respond to questions. In some cases, the longest texts may even approach 200,000 tokens. Typical examples are **LongBench** (Bai et al., 2023), **ZeroSCROLLS** (Shaham et al., 2023), **L-Eval** (An et al., 2023), **BAMBOO** (Dong et al., 2024c), **XL$^2$bench**(Ni et al., 2024), **InfiniteBench** (Zhang et al., 2024b), and **LooGLE** (Li et al., 2023). However, these benchmarks face challenges related to model-specific knowledge requirements, fixed text lengths that are difficult to adjust, and the uncontrollable and subjective nature of content generated by the models.

**Key retrieval**   The other method for evaluating the ability of models to handle extremely long texts involves directly extracting a specified key from unrelated context. This approach is simpler, more flexible, and more controllable. Importantly, it does not rely on the model's existing knowledge. Examples are **Needle in a Haystack** (Kuratov et al., 2024), **Passkey Retrieval** (Mohtashami & Jaggi, 2023), and **RULER** (Hsieh et al., 2024). However, critics argue that this method is overly simplistic.  Examples of these tasks can be found in Appendix B Figure 4 and Figure  5.

**Few-shot Testing**   In addition to datasets specifically designed for long texts, the length of traditional shorter test sets can be extended through a few-shot (Brown et al., 2020) format or by simulating multi-turn dialogues, in order to test the model's capabilities with long texts. However, in long-text tests based on few-shot, texts at a greater distance only serve as a reference and do not contain information that the model needs to extract or conditions for inference, so the reference value of the results is limited.

### 6.2 Evaluation Metric

**Per Token GPU-Memory Usage**   For KV Cache, the most intuitive optimization indicator would be the memory space occupied by each token. The LLaMA2-7B model, as a typical example, theoretically occupies 0.5MB of memory for each KV Cache entry. Please note that when measuring this indicator rigorously, the fragment space generated by the token should also be taken into account, that is, it is best to measure according to the actual memory occupancy, rather than a value calculated through a string of parameters.

**Throughput and Latency**   Throughput and latency are important indicators for measuring the time efficiency of a model. Throughput, usually measured in tokens per second (token/s), represents how many new tokens the model can generate per second. The higher the throughput, the better the model efficiency. In the Decoding phase, latency is usually considered to be the time required to generate each new token, which is the reciprocal of the throughput, typically in milliseconds.

**Perplexity**   The Perplexity (PPL) is: for each token, the model calculates the natural logarithm likelihood value of the probability distribution predicted based on its previous tokens, takes the average, and then calculates the value as the exponent of $e$, mathematically given by Formula 3, where ANLL refers to the average natural logarithm likelihood.

$$\text{ANLL} = -\frac{1}{N} \sum_{i=1}^{N} \log P\left(x_i | x_1 x_2 \ldots x_{i-1}\right), \quad \text{PPL} = e^{\text{ANLL}} \tag{3}$$

PPL can provide a rough reference for the performance changes of the model. If PPL rises sharply, it usually means that the model's ability has significantly decreased, such as completely losing language ability, etc.

## 7 Key Takeaways

This review, following the footsteps of prior work, takes a closer look into KV Cache optimization, uncovering its complexities and proposing strategies for its optimization. We hope to propose some insights and suggested directions for future research that are not just reflections of current trends but also an invitation to explore the uncharted territory of LLMs.

**Principles of KV Cache Optimization**: At the core of optimizing KV Cache lies the principle of reducing memory consumption. This can be achieved by further compressing 'K' (Keys) or 'V' (Values) in the KV pairs. Techniques to compress these components directly impact the efficiency of the models, especially in terms of memory usage and processing speed.

**Trade-offs in Deletion vs. Compression**: Whether to delete less important KV pairs to save memory or to compress the KV Cache without deletion remains an open question. While deletion might offer immediate memory relief, it could potentially compromise the model's performance. In contrast, better compression techniques strive to retain information integrity while reducing memory footprint.

**Extremes in KV Cache Management**: A more radical approach could involve storing the KV Cache externally, possibly on a different storage medium. This method would transform KV Cache management into a retrieval challenge, where the relevant KV pairs are fetched and reintegrated into the model as needed. While this could reduce memory usage on primary devices, it would introduce complexities in retrieval and integration processes.

**Future Directions in Storage and Retrieval Technologies**: These discussions point towards an evolving future where storage and retrieval technologies might become as crucial as the computational models themselves. Innovations in how KV Cache is managed, stored, and accessed could open up new avenues for making LLMs more efficient and versatile.

## 8 Conclusion

In this review, we highlighted the significance and the multifaceted nature of optimizing KV Cache in Large Language Models (LLMs). This review traversed through a variety of methods, from the training and deployment stages to post-training optimizations. Each method, whether it be the architectural changes in the pre-training phase, framework optimizations during deployment, or dynamic strategies like eviction and quantization in the post-training phase, offers unique insights into mitigating the challenges posed by the KV Cache's memory-intensive nature. Moreover, the exploration of different evaluation metrics for long-text performance underpins the importance of a balanced approach that considers both efficiency and model capabilities. As the field continues to evolve, it is clear that the optimization of KV Cache will remain a critical area of focus, offering a pathway towards more efficient and environmentally responsible use of LLMs. We hope this review can be a roadmap to guide learners venturing through this dynamic and rapidly progressing field.

**Acknowledgments**

We sincerely appreciate the valuable feedback provided by all reviewers during the review process, as well as the efforts of the ethics area chairs and program area chairs. This work was supported by the National Natural Science Foundation of China (No. 62306216), the Natural Science Foundation of Hubei Province of China (No. 2023AFB816), the Fundamental Research Funds for the Central Universities (No. 2042023kf0133).

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

## A  Popular Models With GQA

In table 2, we've showcased that how popular models applied GQA/MQA. For each metrics:

- GQA: Indicates whether the model uses the GQA method.
- MoE: Indicates whether the model is a Mixture of Experts model. If yes, it's represented as "Total Experts (Activated Experts)". For example, Mixtral utilized 2 out of 8 experts per token, so it's showcased as 8(2).
- PC: Total parameter count, including parameters from Attention, FFN, LayerNorm or RMSNorm, Embedding, and LM-Head.
- LC: Number of layers in the model.
- HD: Hidden layer dimension. Note that this dimension isn't necessarily the same as the $qkv$ vectors dimension. For example, Gemma-7b uses a $3072x4096$ matrix to map 3072-dimensional $h$ to 4096-dimensional $qkv$.
- $n_h$ and $\bar{n}_q$: Number of attention heads and average number per layer of KV heads. The "average" is necessary here due to the Deci-LM use different $n_q$ value across different layers.
- $S_{HK}$ and $S_{KV}$: Embedding size (in bytes, `bfloat16` as data type) and corresponding KV Cache increment per token.
- F/A: Represents the ratio of parameters in the Feed-Forward Network (FFN) layer to those in the Attention layer. Notably, in models that do not use GQA, this value tends to be around 2. However, in models employing GQA, it typically increases to the range of 3.5 to 5. For MoE models, this is calculated by multiplying the F/A value of single expert with the activate experts counts.
- $\mathcal{R}$: Represents the ratio of the KV Cache size generated for each token to the size of the embedding vector. A higher ratio indicates that the information for this token has been expanded by a greater factor. Notably, when using the GQA method, this value is significantly lower compared to non-GQA methods, implying that the information produced is relatively compressed.

## B  Dataset Examples

For Passkey-retrieval and Haystack, one example is provided for each benchmark, which can be found in Figure 4 and Figure 5.

| Model | GQA | MoE | PC | LC | HD | $n_h$ | $\bar{n}_q$ | $S_{HS}$ | $S_{KV}$ | F/A | $\mathcal{R}$ |
|---|---|---|---|---|---|---|---|---|---|---|---|
| Grok1 | **Yes** | 8(2) | 314B | 64 | 6144 | 48 | 8 | 12K | 163K | 6*2 | 13 |
| DBRX | **Yes** | 16(4) | 132B | 40 | 6144 | 48 | 8 | 12K | 163K | 2.06*4 | 13 |
| Gemma | No | – | 8.5B | 28 | 3072 | 16 | 16 | 6K | 468K | 4.5 | 75 |
| Gemma | **Yes** | – | 2.5B | 18 | 2048 | 8 | 1 | 4K | 18K | 9.6 | 4.5 |
| DeciLM | **Yes** | – | 7.0B | 32 | 4096 | 32 | 2.1 | 8K | 34K | 4.93 | 4.2 |
| DeciLM | **Yes** | – | 5.7B | 32 | 4096 | 8 | 1.6 | 8K | 25K | 3.84 | 3.1 |
| Phi-2 | No | – | 2.8B | 32 | 2560 | 32 | 32 | 5K | 327K | 3 | 64 |
| Deepseek | No | 66(8) | 16.4B | 28 | 2048 | 16 | 16 | 4K | 229K | 1+0.5*6 | 56 |
| Qwen1.5 | **Yes** | – | 72B | 80 | 8192 | 64 | 8 | 16K | 327K | 4.7 | 20 |
| Qwen1.5 | No | 61(5) | 14.3B | 24 | 2048 | 16 | 16 | 4K | 197K | 2+0.5*4 | 48 |
| Qwen1.5 | No | – | 14.2B | 40 | 5120 | 40 | 40 | 10K | 819K | 2.01 | 80 |
| Qwen1.5 | No | – | 7.7B | 32 | 4096 | 32 | 32 | 8K | 524K | 2.01 | 64 |
| Qwen1.5 | No | – | 1.8B | 24 | 2048 | 16 | 16 | 4K | 197K | 2.01 | 48 |
| Yi | **Yes** | – | 34B | 60 | 7168 | 56 | 8 | 14K | 245K | 3.75 | 18 |
| Yi | **Yes** | – | 8.8B | 48 | 4096 | 32 | 4 | 8K | 98K | 3.58 | 12 |
| Yi | **Yes** | – | 6.0B | 32 | 4096 | 32 | 4 | 8K | 65K | 3.58 | 8 |
| Mixtral | **Yes** | 8(2) | 47B | 32 | 4096 | 32 | 8 | 8K | 131K | 4.2*2 | 16 |
| Mistral | **Yes** | – | 7.2B | 32 | 4096 | 32 | 8 | 8K | 131K | 4.2 | 16 |
| GLM2/3 | **Yes** | – | 6B | 28 | 4096 | 32 | 2 | 8K | 28K | 4.72 | 3.5 |
| LLaMA2 | **Yes** | – | 69B | 80 | 8192 | 64 | 8 | 16K | 327K | 4.7 | 20 |
| LLaMA2 | No | – | 13B | 40 | 5120 | 40 | 40 | 10K | 819K | 2.02 | 80 |
| LLaMA2 | No | – | 6.7B | 32 | 4096 | 32 | 32 | 8K | 524K | 2.01 | 64 |

Table 2: The current state of popular open-source LLM using the GQA method, along with their approximate parameter counts.

There is an important info hidden inside a lot of irrelevant text. Find it and memorize them. I will quiz you about the important information there.

**prefix filler**

The grass is green. The sky is blue. The sun is yellow. Here we go.There and back again. (repeat for N times)

The pass key is <PASS KEY> . Remember it. < PASS KEY > is the pass key.

**suffix filler**

The grass is green. The sky is blue. The sun is yellow. Here we go.There and back again. (repeat for M times)

What is the pass key? The pass key is

Figure 4: Prompt format for passkey retrieval. (<PASS KEY> is a 5-digit number.)

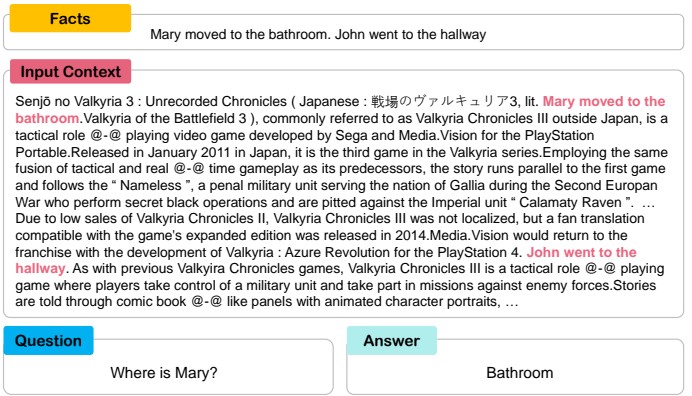

Figure 5: Example for Needle in a Haystack task. Facts are inserted in the sentences of irrelevant text

## C  Other Methods

In addition to the above methods, there are also some methods that modify the Attention mechanism to completely eliminate KV Cache. Furthermore, by directly compressing the prompt or the embedding, the length of KV Cache can also be reduced by decreasing the number of tokens.

### C.1  Linear-Transformers

Katharopoulos et al. (2020); Wang et al. (2020); Verma (2021) proposed a linear attention mechanism. By removing the softmax function and slightly adjusting $W_{\{q,k,v\}}$, the computational complexity of the Attention part in the Transformer is transformed from quadratic to linear, thereby achieving an inference speed consistent with Recurrent Neural Networks (RNNs), and totally eliminate the KV Cache. Other popular Linear-Transformers, mainly RNN-like or SSM models, includes RetNet (Sun et al., 2023b), RWKV (Peng et al., 2023b), Mamba (Gu & Dao, 2024), etc.

Zhang et al. (2024a); Arora et al. (2024) are some improvements to linear attention. These include the use of naive Attention mechanisms at close range, and the optimization of the performance of linear Transformers through Spiky and Monotonic Weights.

### C.2  Prompt and Embedding Engineering

Chevalier et al. (2023) proposed to use summary vectors for compressing history information. It concatenates the summary vectors to long segments and train the model to gather information from previous embedding.

Mu et al. (2023) applies a special attention mask. The gist token is inserted between instructions and inputs, acts as a bridge for transferring instruction information to inputs and thus forcing gist token to compress information.

LLMLingua (Jiang et al., 2023a) uses a budget controller, an iterative token-level compression algorithm and a small model to estimate which token is important. LongLLMLingua (Jiang et al., 2023b) further improves LLMLingua in long context scenarios.

