# OpenReview forum: "Keep the Cost Down: A Review on Methods to Optimize LLM’s KV-Cache Consumption"
_colmweb.org/COLM/2024/Conference — COLM_

### Official Review · Reviewer_XM3o · 2024-05-10

**Rating:** 9
**Confidence:** 4
**Ethics Flag:** 1

**Summary:**

The manuscript presents an up-to-date overview of approaches for optimizing the KV-Cache, which is a method for speeding up, and reducing the memory load for LLMs.

**Questions To Authors:**

Some acronyms are not explained: GQA (page 3), RAG (page 4), please consider explaining.

Page 4: "but each time it only increases a little, which is a horrible property" - the 'horrible' epithet is not helpful. The case is not 'horrible' but surely can be made more efficient.

Page 7, a typo: "obtains the ability to accumulates history" -- should be 'accumulate'

Appendix A: "Mixture of Experts model" - please explain what is that.

Appendix C 'Other methods' deserves to be a section in the body of the paper, and deserves more elaboration. If you get more pages for the paper, consider moving this section.

The following papers in the References section have no venue/publication data:
Arora et al., Beltagy et al., Han et al.
Please check completeness of the references.

**Reasons To Accept:**

A good and comprehensive overview to one of the most important technical aspects of modern LLMs. The introduction is reasonably detailed, and the review is well-structured. Also includes helpful diagrams.
This review can also be used for introducing new researchers to this topic.

Notably, this is a very technical and complicated topic, and covering it within just 9 pages is a feat of compression (pun intended), but the authors also managed to include an evaluation section and neatly describe the logic of KV-Cache evaluations. Well done.

**Reasons To Reject:**

The papers is in some places too condensed.

---

> ### Author Rebuttal · Authors · 2024-05-30
>
> ## W1: The papers is in some places too condensed
>
> Thank you for your careful review. Due to page limitations, we were unable to provide more detailed explanations in some sections. However, we plan to offer more extensive discussions and elaborations in the appendix in our future work. This will ensure that readers can access detailed information without compromising the overall structure and conciseness of the main paper.
>
> ## Q1: Some acronyms are not explained: GQA (page 3), RAG (page 4), please consider explaining
>
> Here GQA refers to  Grouped-query Attention, and RAG refers to Retrieval-augmented Generation.
>
> ## Q2&Q3: Typos and Presentation
>
> Thank you for your suggestion. We will revise and optimize the expressions in the subsequent versions and conduct a more thorough proofreading to correct any typos.
>
> ## Q4: Appendix A: "Mixture of Experts model" - please explain what is that
>
> In the context of transformer models, a MoE(Mixture of Experts) consists of two main elements:
>
> - Sparse MoE layers are used instead of dense feed-forward network (FFN) layers. MoE layers have a certain number of “experts”, where each expert is a neural network. In practice, the experts are FFNs.
> - A gate network or router, that determines which tokens are sent to which expert.
> Please refers to paper (<https://arxiv.org/pdf/2101.03961>) for more detailed explanations.
>
> ## Q5: Appendix C 'Other methods' deserves to be a section in the body of the paper, and deserves more elaboration. If you get more pages for the paper, consider moving this section
>
> Thank you for your invalueable suggestion. We will try to integrate the content from Appendix C into the main body of the paper for a more comprehensive discussion.
>
> ## Q6：The following papers in the References section have no venue/publication data: Arora et al., Beltagy et al., Han et al. Please check completeness of the references
>
> We will double check and revise these references in updated version.

---

> > ### Comment · Reviewer_XM3o · 2024-06-04
> >
> > Thank you for the responses. If the manuscript is accepted, adjust the clarifications in the text.

---

### Official Review · Reviewer_mRjo · 2024-05-10

**Rating:** 3
**Confidence:** 3
**Ethics Flag:** 1

**Summary:**

This paper performs a literature review on optimizing KV-cache usage in LLMs. The paper is well written and ontology on classifying the current research and engineering effort is always welcome. While the topic is an important and active research area, I feel that the paper could have covered a rather deep review and discussion on the topic. The authors didn't discuss financial implications of different approaches while the title says 'Keep the Cost Down'.

**Reasons To Accept:**

The paper focuses important topic that is active and involving. Author attempt to cover approaches more broadly rather than depth.

**Reasons To Reject:**

In depth, discussion and review is missing since it is a survey paper. Work could be supplemented by experimental benchmarks.

---

> ### Author Rebuttal · Authors · 2024-05-30
>
> ## W1: In depth discussion and review is missing since it is a survey paper. Work could be supplemented by experimental benchmarks
>
> Thank you for your feedback. We have included an in-depth discussion review of the different methods explored in the Key Takeaways section. Besides, we have also provided our insightful perspective on this critical problem and pointed out promising potential future directions for the research.
>
> Moreover, we've conducted a needle-in-a-haystack test on the methods mentioned in the Post-Training Optimization section, as most of them did not test this in their paper. We also planned to add a section of the pros and cons of various methods in section 5. Results are as follows.
>
> | Scenarios / Property | Quantization | Dynamic Eviction | Merging |Static Eviction |
> |-|-|-|-|-|
> | Short context | ✓ | ✓ | ✓ | ✓ |
> | Long context generation | ✓ | ✓ | ✓ | ✓ |
> | Long context retrieval | ✓ | ✗ | ✓ | ✗ |
> | Time efficiency | ○ | ○ | ○ |  ✓ |
> | No extra training | ✓ | ✓ | ○ | ✓ |
> | Flash-Attn friendly | ✓ | ○ | ✗ | ✓ |
> | CPU friendly | ✗ | ✓ | ✓ | ✓ |
> | Needle-in-haystack | ✓ | ✗ | ✓* | ✗ |
>
> ✓ and ✗ refers to suitable and not suitable. ○ refers to partially suitable or depends on the implementation. \* refers to results reported within the paper. The ‘Merge’ in the table refers to a type of method that emerged in early April 2024, which merges the KV Cache of different tokens. This is a part that we added when we continued to maintain this work.
>
> ## W2: The authors didn't discuss financial implications of different approaches while the title says 'Keep the Cost Down'
>
> Thank you for suggesting another possible interpretation of our title. But here, our ‘cost’ is more inclined towards the consumption of performance, specifically in terms of memory, I/O bandwidth, and computational resources. While the KV Cache does indeed impact the financial cost of the model, the financial cost is also influenced by many factors unrelated to language models and subject to rapid changes. These are irrelevant to the issues we are concerned with, so we have not discussed this aspect.

---

> > ### Comment · Reviewer_mRjo · 2024-06-04
> >
> > Thanks for taking extra effort for the Needle-in-haystack eval! It is nice have such eval and is useful to the community. I am willing to increase my score to 5.

---

### Official Review · Reviewer_dhyD · 2024-05-11

**Rating:** 6
**Confidence:** 3
**Ethics Flag:** 1

**Summary:**

This paper provides a review of KV-cache optimization for LLMs. The reviews are grouped according to the stage when cache optimization is performed: training, deployment or post-training. Training-stage optimization mainly changes the model's architecture for efficient caching, deployment-stage optimization mainly studies better inference system, and post-training optimization mainly perform token eviction or quantization. Moreover, related datasets and evaluation metrics are also discussed.

**Reasons To Accept:**

- Techniques on KV-cache compression are crucial for efficient deployment of LLMs.
- A nice review is provided by this work, which may inspire more future research on this direction.
- This work provides a good categorization of approaches on KV-cache compression, classified into training-stage, deployment-stage and post-training stage, which clearly describes how and when we can optimize the KV caches.

**Reasons To Reject:**

- This paper is mainly a literature review without empirical experiments exploring which methods are most effective in which scenarios.
- It would be nice to provide a table listing the advantages and disadvantages of different methods with suggestions on their best usage.
- Prompt compression techniques are quite related to this topic and related work should be discussed.

---

> ### Author Rebuttal · Authors · 2024-05-30
>
> ## W1: This paper is mainly a literature review without empirical experiments exploring which methods are most effective in which scenarios
>
> Thank you for your careful review. We would like to clarify that in the first paragraph of the second page, we divided the methods into pre-training, post-training, and deployment stages, and discussed the appropriate scenarios for each method.
> For those with sufficient computational resources, methods like MQA and GQA are recommended. As listed in Appendix A, many organizations that can train a large language model (LLM) from scratch have successfully used GQA and MQA. For those with limited resources, methods in the post-training and deployment stages are more suitable.
> During the rebuttal period, we also created supplementary tables to help readers gain a deeper understanding of the different KV Cache optimization methods in next section.
>
> ## W2: It would be nice to provide a table listing the advantages and disadvantages of different methods with suggestions on their best usage
>
> Thank you for your suggestion. During the rebuttal period, we have added a supplementary table to compare different methods, highlighting their advantages and disadvantages along with suggestions for their best usage.
>
> | Scenarios / Property | Quantization | Dynamic Eviction | Merging |Static Eviction |
> |-|-|-|-|-|
> | Short context | ✓ | ✓ | ✓ | ✓ |
> | Long context generation | ✓ | ✓ | ✓ | ✓ |
> | Long context retrieval | ✓ | ✗ | ✓ | ✗ |
> | Time efficiency | ○ | ○ | ○ |  ✓ |
> | No extra training | ✓ | ✓ | ○ | ✓ |
> | Flash-Attn friendly | ✓ | ○ | ✗ | ✓ |
> | CPU friendly | ✗ | ✓ | ✓ | ✓ |
> | Needle-in-haystack | ✓ | ✗ | ✓* | ✗ |
>
> ✓ and ✗ refers to suitable and not suitable. ○ refers to partially suitable or depends on the implementation. \* refers to results reported within the paper. The ‘Merge’ in the table refers to a type of method that emerged in early April 2024, which merges the KV Cache of different tokens. This is a part that we added when we continued to maintain this work.
>
> ## W3: Prompt compression techniques are quite related to this topic and related work should be discussed
>
> Please refer to Appendix C where we have included a brief introduction of prompt, or embedding, compression techniques and related work. Besides, we will integrate the content from Appendix C into the main body of the paper for a more comprehensive discussion.

---

> > ### Comment · Reviewer_dhyD · 2024-06-05
> >
> > Thanks for your response and please update the paper in later versions accordingly.

---

### Official Review · Reviewer_q2wL · 2024-05-13

**Rating:** 5
**Confidence:** 3
**Ethics Flag:** 1

**Summary:**

This submission is a review of the compression methods of KV Cache. It organizes existing works in chronological order, including the training phase, deployment phase, and run-time phase with post-training compression. The training stage mainly involves some novel architecture designs, such as Multi-Query Attention (MQA), and Grouped-Query Attention (GQA). For the deployment-related techniques, this paper covers some system optimization like Paged Attention and DistKD-LLM. For the post-training optimization, several training-free methods are discussed, including Window Attention and Attention Sink. Besides, some pruning, and quantization methods are mentioned. Further, this paper summarises the datasets and evaluation metrics for the KV cache.

**Questions To Authors:**

Please refer to the weaknesses. My main concern lies in the lack of in-depth analysis of similar methods.

**Reasons To Accept:**

1. The review's structure is clear, covering multiple stages including pre-training, deployment, and run-time optimization, facilitating a comprehensive understanding.
2. Both efficiency and model capacity, crucial aspects for KVCache, are thoroughly discussed, demonstrating a holistic approach to optimization.
3. The paper provides four key takeaways for KV Cache optimization, Principles of KV-Cache Optimization, Trade-offs in Deletion vs. Compression, Extremes in KV-Cache Management and Future Directions in Storage and Retrieval Technologies. These insights offer valuable guidance for future research, highlighting important areas for further exploration and advancement.

**Reasons To Reject:**

1. The primary reason for rejection is the lack of connections between different methods within the paper. While the taxonomy is structured around different stages, such as pre-training, deployment, and run-time optimization, there is a missed opportunity to highlight similarities and shared concepts among various methods. For instance, the Attention Sink and Repetitive Attention Pattern both emphasize the importance of focusing on specific tokens, yet this connection is not adequately explored or summarized in the review. As a review paper, it's crucial to provide a synthesis of such related ideas to enhance clarity and coherence.
2. Despite claiming to introduce metrics for assessing LLMs' performance on long texts, the evaluation section falls short by only listing general and widely used metrics like memory usage, throughput, latency, and perplexity. The paper lacks a comprehensive assessment of different methods, failing to delve into specific metrics tailored to the nuances of long text processing. A more thorough evaluation framework is needed to effectively gauge the effectiveness of the reviewed methods and provide valuable insights for future research and development in this domain.

---

> ### Author Rebuttal · Authors · 2024-05-30
>
> ## W1: Lack of connections
>
> Thank you for your suggestion. In fact, in Section 5.1, paragraph 5, we have discussed in detail how different works estimate the future significance of a specific token based on the historical information of attention weights, exploring their similarities and differences. This paper focuses on categorizing and summarizing different methods in detail. For eviction methods, we have decomposed them into two lines of approaches: static policies and dynamic policies, based on whether they use manual or dynamic methods to evict unnecessary tokens.
>
> We will include the discussion of Attention Sink and Repetitive Attention Pattern in future versions and establish connections between different works within each category to enhance clarity and coherence. Besides, we will seek for more connections between methods. For example, methods like GQA and Eviction both focused on how to reduce the number of stored variables, making the information more condensed.
>
> ## W2: The evaluation section falls short
> Thank you for your suggestion. We have already listed the mainstream methods for testing ultra-long context capabilities in Section 6.1. Nevertheless, our current chapter layout indeed occupies too much space on common metrics that are less related to big KV Cache scenarios. In the next version, we will focus more on supplementing benches focused on long-text, since they are more relevant to evaluate the KV Cache compression. We will also add discussions on the current shortcomings of benches. The main content is an unresolved contradictions, that is, evaluation methods with long texts are often simple in form and task, due to the sparsity of critical information; evaluation methods that are complex in form and difficult in task are difficult to extend to sufficient lengths.
>
> Moreover, for methods mentioned in section 5, we provide a table describes the pros and cons of them.
> |Scenarios/Property|Quantization|Dynamic Eviction|Merging|Static Eviction|
> |-|-|-|-|-|
> |Short context|✓|✓|✓|✓|
> |Long context generation|✓|✓|✓|✓|
> |Long context retrieval|✓|✗|✓|✗|
> |Time efficiency|○|○|○|✓|
> |No extra training|✓|✓|○|✓|
> |Flash-Attn friendly|✓|○|✗|✓|
> |CPU friendly|✗|✓|✓|✓|
> |Needle-in-haystack|✓|✗|✓*|✗|
>
> ✓ and ✗ refers to suitable or not. ○ refers to partially suitable or depends on the implementation. \* means result reported by the paper. The ‘Merge’ in the table refers to a type of method that emerged in early April 2024, which merges different tokens' cache.

---

### Official Review · Reviewer_uKZX · 2024-05-23

**Rating:** 6
**Confidence:** 3
**Ethics Flag:** 1

**Summary:**

This paper presents a review of KV-caching techniques for Decoder-only LLMs, describing methods and strategies pertaining to multiple steps along the life of a model, as well as briefly discussing evaluation metrics and proposing avenues for future work. Overall, the work is timely and necessary, and there will definitely be benefits for researchers new to this topic who choose to read it instead of (or before) diving into the growing literature.

**Questions To Authors:**

### Questions
* In S2, does $p_i$ denote a distribution, as suggested twice, or a single probability, as suggested once and inferred by convention?
* In S3, how do $n_h$ combination weights result in $n_h$ combinations? It feels like an entire dimension is absent in the former.
* In S3, setting $n_g$ to $n_h$ in the $\eta$ equation results in 1.5, which cannot be true (it needs to be 1). What's the correct formula?

### Required fixes
* Please refrain from personifying models - "understanding" in line 4 of the intro should be changed. Especially since this is a didactic work which may be used in introductory classes.

### Grammar and Suggestions
* There are some foci of grammatical errors - four in the last three lines of the first intro paragraph, the caption for figure 2, "utilize(s)"in p.5, "quer(y/ies)" in p.6. The last sentence of the intro is not understandable.
* Appendices A and B look useful, you should probably alert the reader to their existence in the main text.
* The transpose notation in eq. (1) and (2) is not the standard "top" but a cross, please change.
* Please don't call things "horrible" (top of S4). "undesirable" would be more suitable.
* In p.5, *Static policies* paragraph, it seems like Xiao et al. and Han et al. *find* rather than *suggest* (unless they really only suggest, in which case somebody better run that analysis).
* Last but not least. "(the) KV cache" is a noun phrase and shouldn't be hyphenated unless used as a modifier.

**Reasons To Accept:**

* The topic is relevant and in need of a review.
* The paper can be of use to both practitioners wondering which strategy to pursue in the model they're training and for newcomers who want to design the next shiny KV caching technique.
* The method-focused core of the review (sections 3--5) is well-structured and well-balanced among the importance and "meat" in the various sub-sub-topics. The introduction is also good.

**Reasons To Reject:**

* The non-core sections are substantially weaker. "Evaluation" (S.6) is mostly unnecessary and the space used for it could have been better spent on covering more methods, extending the descriptions for some, or integrating interesting bits from Appendices, mostly C. Not only is S6 describing at length well-known resources, it does not connect back to any of the methods described in the core sections, and the results table isn't referred to, let alone analyzed. This is followed by a "take-aways" section (S7) which does not follow comfortably from the preceding ones. Overall, there's a noticeable disconnect at the 5-6 section border.
* Section 2 (Preliminaries and Notation(s)) is a good idea but not executed carefully enough (see below).
* While some illustrations (Fig. 2,3) are useful and illustratory, Figure 1 is mostly confusing and inconsistent. It does not lend intuition of the topic or set expectations for the survey's contents.

---

> ### Author Rebuttal · Authors · 2024-05-30
>
> ## W1: The non-core sections are substantially weaker. There's a noticeable disconnect at the 5-6 section border
>
> Thank you for your valuable feedback. The primary goal of Section 6 was to provide beginners with an understanding of the commonly used datasets and metrics for KV Cache optimization. In the next version of our paper, we will restructure the content by reducing and moving Section 6 to the appendix. Additionally, we will offer a more detailed discussion of the results table and integrate the content from Appendix C into the main body of the paper. We will also enhance the logical flow between sections.
>
> ## W2&Q1: Does $p_i$ denote a distribution or a single probability
>
> As stated in Section 2, $p_i \in \mathbb{R}^V$ denotes a distribution, not a single probability.
>
> ## Q2: how do $𝑛_ℎ$ combination weights result in $𝑛_ℎ$ combinations?
>
> As explained in section 3 in page 3, in MQA, all heads share the same Key and Value matrices, while each head retains its own set of Query parameters. Therefore, for $n_h$ heads, there are $n_h*1=n_h$ different QKV combinations.
>
> ## Q3: In Section 3, what's the correct formula?
>
> We apologize for the typo. The correct formula is:
> $$
> \eta = \frac 12 \cdot \left( 1 + \frac{n_g}{n_h} \right)
> $$
> where the weight of the queries and the output projection layer remains unchanged, but the Key and Value matrices are reduced to \($\frac{n_g}{n_h}$\) of their original size.
>
> ## W3 :Some illustrations is confusing and inconsistent
>
> Thank you for your feedback. The purpose of Figure 1 is to provide a roadmap to help readers gain a clearer understanding of the overall structure of the paper. Our paper unfolds in a chronological order of LLM: (1) In the training stage, we introduce the KV Cache compression methods that can be used during model pre-training. These methods are usually the most effective, but are not suitable for modifying existing models with low computational power. (2) In the deployment stage, we discuss the use of different frameworks to optimize the use of KV Cache which do not require significant modifications to the KV Cache itself but can significantly enhance its efficiency. (3) In the post-training stage, we introduce a large number of on-time optimization methods for KV Cache.
>
> We will improve the illustrations to provide more clarity and detailed explanations.
>
> ## W4: Grammar
>
> We will thoroughly optimize and proofread the presentation to address the grammatical errors.

---

### Decision · Program_Chairs · 2024-07-10

**Decision:**

Accept

**Comment:**

The reviewers agree this paper is on a timely topic and seems to be very thorough, especially for a conference paper. The authors have promised to add some additional discussion and synthesize their different threads more closely, including adding a table to compare the methods in question. If they make these changes, the paper is clearly strong enough to accept.

[At least one review was discounted during the decision process due to quality]